# Effects of Dietary Hemp Seed and Flaxseed on Growth Performance, Meat Fatty Acid Compositions, Liver Tocopherol Concentration and Bone Strength of Cockerels

**DOI:** 10.3390/ani10030458

**Published:** 2020-03-10

**Authors:** Miloš Skřivan, Michaela Englmaierová, Tomáš Taubner, Eva Skřivanová

**Affiliations:** 1Department of Nutrition Physiology and Animal Product Quality, Institute of Animal Science, 104 00 Prague-Uhrineves, Czech Republic; englmaierova.michaela@vuzv.cz (M.E.); taubner.tomas@vuzv.cz (T.T.); skrivanova.eva@vuzv.cz (E.S.); 2Department of Microbiology, Nutrition and Dietetics, Czech University of Life Sciences Prague, 165 00 Prague Suchdol, Czech Republic

**Keywords:** broiler chickens, breast muscle quality, vitamin E, n-3 polyunsaturated fatty acid, tibia minerals

## Abstract

**Simple Summary:**

Different combinations of extruded flaxseed (EF; 0 and 60 g/kg) and hemp seed (HS; 0, 30, 40 and 50 g/kg) in the cockerel diet are compared. Rapeseed oil is used to balance the energy in the control diet without HS and EF supplementation. The diets are provided ad libitum throughout the fattening period (35 days). Cockerel body weight, feed intake, feed conversion, vitamin E deposition and meat and bone quality are evaluated. The highest body weight at both 14 (511 g) and 35 days of age (2493 g) and the lowest n-6/n-3 fatty acid ratio in breast meat (1.75) is observed in the cockerels fed 60 g/kg EF and 40 g/kg HS. The combination of the highest levels of HS (50 g/kg) with 60 g/kg EF increases the vitamin E content in the liver compared to diets supplemented with HS or EF only. The hemp seed alone (346.3 N) and the combination of HS and EF (40 g/kg HS with 60 g/kg EF (359.2 N) or 50 g/kg HS with 60 g/kg EF (358.3 N)) positively influences the bone strength in the cockerels compared to the control group (297.3 N), a fact that could result in a reduction in fracture incidence and an improvement in breeding efficiency.

**Abstract:**

The aim of the study is to determine the effect of hemp seed (HS) of the nonpsychotropic variety Futura and extruded flaxseed (EF) in the diet of cockerels on cockerel growth performance, breast muscle and liver α- and γ-tocopherol concentrations, breast muscle fatty acid concentrations and tibia strength. Five hundred and forty one-day-old male Ross 308 cockerels are equally allocated into six groups. Each group has three replicates of 30 cockerels in pens with litter. The formulated diets are isoenergetic (the metabolisable energy ranged from 12.4 to 12.8 MJ/kg) and isonitrogenic (the protein concentration ranged from 209.7 to 210.9 g/kg) and provided ad libitum. During the experiment, which lasts 35 days, the control group is fed a diet without EF or HS. Rapeseed oil was the lipid source in the control diet. The diet for the second group contains EF at 60 g/kg, the diet for the third group contains HS at 40 g/kg, and the diets for the fourth to sixth groups contain HS and EF at 30 and 60 g/kg, 40 and 60 g/kg and 50 and 60 g/kg, respectively. At the end of the experiment, 15 cockerels of average weight are slaughtered per group, and the breast muscle, liver and tibia bone are dissected for chemical analyses. The all dietary combination of HS and EF increases (*p* < 0.001) cockerel body weight (2375–2493 g) more than HS alone (2174 g) or EF alone (2254 g). A similar finding is observed for the diet composition and tocopherol content in the liver, but the doses of HS required to achieve this effect are higher (40 and 50 g/kg). The tocopherol content in the breast muscle is not influenced by the diet. The dietary combination of 60 g/kg EF and 40 g/kg HS results in the most promising findings of the experiment, since it leads to the lowest n-6/n-3 polyunsaturated fatty acids ratio (*p* < 0.001; 1.75). Incorporation of HS into the diet increases cockerel tibia strength (*p* < 0.001), which is of great practical importance due to the frequent occurrence of limb fractures. To conclude, the dietary supplementation with 40 g/kg HS and 60 g/kg EF improves cockerel performance, meat and bone quality and deposition of α-tocopherol in the liver.

## 1. Introduction

Flaxseed (*Linum usitatissimum*) is appreciated for the favourable representation of n-3 fatty acids, especially α-linolenic acid (ALA). Flaxseed contains 35–45% oil of which 45–52% is ALA [1], which is a precursor of the other two n-3 fatty acids, eicosapentaenoic acid (EPA) and docosahexaenoic acid (DHA) [2]. Regarding humans, the n-3 fatty acids reduce the risk of coronary heart disease, osteoporosis, rheumatoid arthritis, autoimmune disorders or cancer [3]. Therefore, according to recommendations of the World Health Organization, the n-6/n-3 polyunsaturated fatty acid (PUFA) ratio should be 4–5 or less. The inclusion of flaxseed in the chicken diets significantly increases the ALA in the breast tissue with no difference in the organoleptic quality of the meat [4]. Higher content of unsaturated fatty acids in chicken meat leads to higher susceptibility to oxidation, therefore it is necessary to ensure a sufficient dose of antioxidants, e.g., vitamin E, in the feed mixture.

Hemp seed (HS, *Cannabis sativa*) cultivars with low tetrahydrocannabinol concentrations are a desirable source of bioactive substances. Hemp oil is also an important source of tocopherols, which can reduce cardiovascular disease, cancer and age-related macular degeneration [5]. The variability in oil, tocopherol and fatty acid levels in hemp varieties was described by Kriese et al. [6], who examined seeds of 51 hemp genotypes from two consecutive harvests. The main tocopherol in hemp seeds is γ-tocopherol, followed by α-, δ- and β-tocopherols. The contents of these tocopherols (γ-, α-, δ-, and β-) in seeds of 51 genotypes averaged 21.68, 1.82, 1.20 and 0.16 mg/100 g, respectively. These findings are consistent with the study of Oomah et al. [7]. Due to the high levels of unsaturated fatty acids (FAs) in HS, tocopherols act as antioxidants to prevent the oxidation of these FAs [8]. α-Tocopherol is generally considered to be more effective than γ-tocopherol as a chain-breaking antioxidant that inhibits lipid peroxidation. Cooney et al. [9] suggested that the unsubstituted position of C-5 gamma-tocopherol enables the most efficient capture of lipophilic electrophiles, such as reactive nitrogen species. This hypothesis has been repeatedly confirmed [10]. Both dietary α-tocopherol and γ-tocopherol are largely transferred into egg yolks when HS is present in hen diets [11]. However, it is unclear whether γ-tocopherol from the HS and extruded flaxseed (EF) can also be transferred to the muscles of chickens.

A nonpsychotropic component in HS, cannabidiol (CBD), is important for human and animal nutrition. A positive effect of CBD on the treatment of fractures has been observed due to the resulting increase in collagen metabolism [12]. Sazmand et al. [13] showed a positive effect of *Cannabis sativa* extract on the morphology and growth of bone marrow mesenchymal stem cells in rats. Fractures in chickens cause problems in all indoor poultry systems. Regarding the case of hens, the sternum is fractured most often; in chickens, limb fractures reduce the economy of breeding. Dietary supplementation with HS increases tibia strength in hens [11].

To date, studies on the effects of the addition of HS or hemp oil to chicken and hen feed have been focused on performance and the fatty acid (FA) concentrations in eggs and meat [14,15,16,17,18]. According to published data, the amount of HS used in the chicken diet was 2.5–30%. Khan et al. [14] tested the effects of HS incorporation at the level of 5–20% in chickens and found that at 20% HS, the chickens had the best weight gain and most efficient feed conversion. Eriksson and Wall [15] obtained positive results with higher doses of HS cake in organic chicken breeding. Mahmoudi et al. [18] studied the effects of 2.5–7.5% HS in diets and obtained inconsistent results. The addition of the lowest concentration of HS, 2.5%, led to reduced weight gain and feed intake. To contrast, the addition of three-fold that amount of HS, 7.5%, did not reduce weight gain or intake levels and, in addition, it decreased the levels of triglycerides, low-density lipoprotein cholesterol and total serum cholesterol and increased the amount of high-density lipoprotein cholesterol. Additionally, HS has a high fibre content (28%).

Polyunsaturated fatty acids (PUFAs) are predominant in hemp oil, in which the main FA is linoleic acid (LA), which constitutes 60% of all FAs. α-Linolenic acid (ALA) comprises 17–19% of the total FAs [19], and the ratio of LA (n-6) to ALA (n-3) is approximately 3.3. To reduce this ratio in the diet, incorporation of plant materials, such as flaxseed, which has a high ALA content and is suitable for enriching chicken meat with n-3 PUFAs, is primarily considered [20]. However, antinutritional factors in flax may reduce chicken performance [21]. The concentration of antinutritional substances in flaxseed can be greatly reduced by extrusion [22]. Chicken meat could be a potential sustainable source of EPA and DHA in human nutrition, provided these FAs could be synthesized from plant-derived ALA. Crespo and Esteve–Garcia [23] showed higher rates of de novo FA synthesis in chickens fed a diet supplemented with linseed oil compared with those fed tallow or sunflower oil. Moreover, chickens may be able to metabolise more docosapentaenoic acid (DPA) through tetracosapentaenoic acid, the precursor of DHA, than other species such as rats [24].

The aim of this study is to determine the effect of HS of the nonpsychotropic variety Futura and EF in the cockerel diet on the performance, breast muscle and liver tocopherol concentrations, breast muscle FA concentrations and tibia strength of cockerels.

## 2. Materials and Methods

A total of 540 one-day-old Ross 308 cockerels were used in this experiment. The cockerels were randomly assigned to six treatments with 3 replicate pens (30 cockerels per pen) in each treatment according to the content of HS and EF (Table 1) in the diet (Table 2). The diets were formulated as isoenergetic (the metabolisable energy ranged from 12.4 to 12.8 MJ/kg) and isonitrogenic (the protein concentration ranged from 209.7 to 210.9 g/kg). A balanced metabolisable energy concentration in the diet was achieved by adjusting the amount of rapeseed oil added. The control diet had rapeseed oil as the oil source and did not contain HS or EF. The second and third group of cockerels were fed diets containing 60 g/kg EF and 40 g/kg HS, respectively. The cockerels from the fourth, fifth and sixth group received a combination of the above seeds at HS/EF ratios of 30/60, 40/60 or 50/60. The level of rapeseed oil in the experimental diets was reduced in comparison with the control. Concerning the case of HS, the nonpsychotropic variety Futura was used. The amount of HS used in the present study was based on the results of an experiment by Skřivan et al. [10], who had used 3, 6 and 9% HS of the Futura genotype in a hen diet and achieved the best results at 3%. Therefore, the concentrations of 3, 4 and 5% HS were selected. The EF contained 70% flaxseed and 30% wheat bran, which provided the fat material and fat absorbent material in the extrusion. The amount of flaxseed alone in the EF was 70%, i.e., 42 g of the total 60 g, a proportion that is almost identical to the dietary average of 40 g of HS. The composition and nutrient analyses of the diets are shown in Tables 1, 2, 3 and 5. Shown in Table 3, the FA concentrations in the HS diet showed higher total concentrations of saturated fatty acids (SFAs), monounsaturated fatty acids (MUFAs) and PUFAs than those in the diet with EF alone, with the exception of n-3 PUFAs. Feed and water were provided ad libitum. Cockerels were kept in pens on wood shavings (13.5 cockerels per m^2^). The room was heated with a gas heating source and ventilation was provided with a temperature-controlled fan. Each pen was equipped with nipple drinkers and pan feeders. The environmental conditions were kept in accordance with the requirements for cockerels. The lighting programme from 1 to 7 days was 23 h light and 1 h dark and then 16 h light and 8 h dark. The cockerels were weighed at 0, 14 and 35 days of age. The average body weight of the cockerels at the beginning of the experiment ranged from 43.2 to 43.4 g per group, and no significant differences among the groups were noted (Table 4). Health status was checked twice a day during the experiment based on chicken activity, normal behaviour patterns (e.g., active feed and water intake, normal walking, wing stretching, calm and effortless breathing, energetic movements when distracted), voice, skin, plumage quality, stance and foot and limb formation. Feed intake was monitored daily on a per-pen basis. Feed conversion was calculated as the total feed intake divided by the total weight gain over 35 days.

At 35 days of age, 15 cockerels (5 cockerels from each pen, 5 × 3) with an average body weight were selected from each dietary treatment group and slaughtered. Their carcasses were cooled for 24 h, and then the breast muscle, liver and a tibia bone were dissected for chemical analyses. Procedures performed with the animals were in accordance with the Ethics Committee of the Central Commission for Animal Welfare at the Ministry of Agriculture of the Czech Republic (Prague, Czech Republic) and carried out in accordance with Directive 2010/63/EU for animal experiments. The protocol of this experiment was approved by the Ethical Committee of the Institute of Animal Science (Prague–Uhříněves, Czech Republic), case number 06/2018.

### 2.1. Analyses

Nutrient analyses of the diets, HS and EF were performed according to Association of Official Analytical Chemists (AOAC) methods [25]. The dry matter of the diets, HS and EF was determined by drying to constant weight at 105 °C in an oven (Memmert ULM 500; Memmert, Schwabach, Germany). The crude protein content was measured using a Kjeltec Auto 1030 instrument (Tecator, Höganäs, Sweden). The fat content was determined upon extraction with petroleum ether using a 1045 Soxtec Tecator extraction system (Tecator). Dry homogenized diets, HS and EF were heated to 550 °C (muffle furnace LMH 11/12 with Ht40 AL temperature controller, LAC Asia Limited, Hong Kong, China) and the mineral ash dissolved in 3 M hydrochloric acid. The total phosphorus in the solution was determined using vanadate-molybdate reagent (AOAC International, 2005; method No. 965.17). The calcium concentration in the hydrochloric acid extract was measured by atomic absorption spectrometry using a Solaar M6 instrument (TJA Solutions, Cambridge, UK).

The concentrations of α-tocopherol and γ-tocopherol in the diets, HS, EF, breast meat and liver were determined after saponification and diethyl ether extraction in accordance with the EN 12822 (European) standards [26].

The FA profiles of the diets, HS, EF and breast meat were determined after chloroform-methanol extraction of the total lipids [27]. The alkaline trans-methylation of the FAs was performed as described by Raes et al. [28]. The fatty acid methyl esters (FAME) were discerned by gas chromatography using an HP 6890 chromatograph (Agilent Technologies, Inc., Santa Clara, CA, USA) with a programmed 60 m DB-23 capillary column (150–230 °C) and a flame-ionization detector; the split injections were performed using an Agilent autosampler. One-microliter samples of FAME in hexane were injected at a 1:40 split ratio. Separation was achieved using the following column temperature programme: initially, the column was operated at 60 °C for 7 min, and then, the temperature was programmed to increase by 20 °C/min to 110 °C, which was maintained for 4 min; then, the temperature was programmed to increase by 10 °C/min to 120 °C, which was maintained for 4 min; then, the temperature was programmed to increase by 15 °C/min to 170 °C and by 2 °C/min to 210 °C, which was maintained for 13.5 min and, finally, the temperature was programmed to increase by 40 °C/min to 230 °C, which was maintained for 7 min. The FAs were identified by their retention times compared with standards: PUFA 1, PUFA 2, PUFA 3 and a 37-component FAME mix (Supelco, Bellefonte, PA, USA).

The tibia bones were excised from the carcasses and cleaned of all tissue. The strength required to break the tibia and deformation of the tibias were measured using an Instron 3342 apparatus (Instron) with a 50-kg load cell with a crosshead speed of 50 mm/min. Each tibia was supported on a 5.75-cm span. Then, the bones were dried at 105 °C for 72 h, placed in a desiccator, and weighed to determine the dry weight. Then, the bones were homogenized and placed in a muffle furnace at 600 °C for 24 h, cooled in a desiccator, and the ash weight was recorded. The phosphorus and calcium contents of the tibia ash were determined using the same methods as previously described. The magnesium concentration was measured in the hydrochloric acid extract by atomic absorption spectrometry using a ContrAA 700 F instrument (Analytik Jena AG, Jena, Germany).

### 2.2. Statistical Analyses

The data from the experiments were analysed using analysis of variance (ANOVA) with a general linear model (GLM) procedure in SAS software (Statistical Analysis Software, Version 9.3) [29]. One-way ANOVA was used for comparisons. The pen was the experimental unit (*n* = 3). Differences were considered significant at *p* < 0.05. The results in the tables are presented as the means and the standard errors of the means (SEMs). 

## 3. Results

The cockerels fed the dietary combination of HS and EF had a higher (*p* < 0.001) body weight at both 14 and 35 days of age than the cockerels fed with HS and EF alone or the control diet (Table 4). The same pattern was found for feed intake, but differences in the results from feed comparisons showed a lower level of significance (*p* < 0.05). Feed conversion and mortality were not influenced by dietary treatment.

The concentration of tocopherols in the breast meat was not influenced by HS or/and EF dietary supplementation. The concentrations of α-tocopherol (*p* = 0.017) and γ-tocopherol (*p* = 0.028) in the liver were influenced by the dietary treatment (Table 5).

Evident from Table 6, significant differences in monounsaturated fatty acid (MUFA) levels in breast meat (*p* = 0.011) were found, with greater levels found in the control. Regarding n-3 PUFAs (*p* = 0.011), a higher level was determined with the diet combination of 40 g/kg HS and 60 g/kg EF (Table 6). The breast meat from cockerels fed the HS/EF supplemented diet at the level of 40 and 60 g/kg, respectively, had the lowest ratio of n-6/n-3 FAs (1.75) (*p* < 0.001) and an increased n-3 long-chain PUFA concentration. The concentration of eicosatrienoic acid 20:3 n-3 (ETE) was also higher in this group (*p* < 0.001) than in cockerels fed EF or HS alone. Additionally, the eicosatetraenoic acid 20:4 n-3 (ETA) concentration was also increased (*p* = 0.050) in this group. The EPA 20:5 n-3 content in the breast meat of the cockerels from the fifth group exceeded the concentration of this FA in the meat of the cockerels fed HS alone or a lower concentration of HS in combination with EF (*p* < 0.001). Additionally, in this group, the DPA 22:5 n-3 FA content was the highest (*p* = 0.002) compared to the content obtained with all other dietary treatments, and the DHA 22:6 n-3 FA in the breast muscle was at a higher (*p* = 0.013) concentration than that achieved with the HS treatment alone and was consistent with that of the EF treatment alone. There was no significant difference in the content of n-3 FAs in the combined HS and EF supplemented diets. Linoleic acid 18:2 n-6 and other n-6 FAs in the breast muscle did not change in cockerels fed a dietary combination of HS and EF, but the concentration of n-3 PUFAs was increased. The fat concentration in the breast meat was 1%, with no significant differences among the treatment groups.

Hemp seed alone and the combination of HS (40 and 50 g/kg) and EF (60 g/kg) in the diet increased tibia strength (*p* < 0.001) compared to the control diet (Table 7). A lower degree of tibia deformation was found in all experimental treatments (*p* < 0.001) than in the control, as determined by the instrumental measurements. There were no significant differences in the ash, calcium, magnesium and phosphorus contents of tibias.

## 4. Discussion

According to the existing literature, the results from published experiments on dietary HS supplementation in poultry have consistently shown its positive effect on performance and product quality [14,15,16,17,18]. The experiments conducted by these authors differed in the amount of HS in the diets, but the levels used were often higher than 20%. The high fibre content of HS, up to 30%, does not appear to be an obstacle to its use, although it is known that excessive dietary fibre reduces nutrient digestibility. Contrasting to the majority of the published data, the dietary component of HS in our experiment is low but effective. Exceeding the HS limit of 30 g/kg led to a reduction in the performance of hens in a previous study [11]. Accordingly, in the present cockerel experiment, the best results are achieved with 40 g/kg of HS in a dietary combination with 60 g/kg of EF. Antinutritional agents, especially polyphenols and phytate, are a limiting factor for the use of HS in the diet [30]. A greater negative effect of antinutritional agents is found for EF than for HS. It has already been shown that 5% EF reduced the performance of chickens [4]. The decrease in broiler growth with diets containing flaxseed may be due to the presence of mucilage, cyanogenic glycosides, and allergens and vitamin B6 antagonism [31]. To contrast, the extruded EF used in this experiment increases the body weight of the cockerels at 35 days compared to the control without EF and without HS. The oil, vitamin E and FA contents also vary according to the genotype of the hemp. The data reported by Kriese et al. [6] on fat and vitamin E concentrations in Futura HS were very close to the results of our analyses. Hemp seed contains high levels of γ-tocopherol, which was until recently neglected in human nutrition [32]. While HS and EF do not increase the concentration of α-tocopherol in the diets, the concentration of γ-tocopherol is higher due to the HS. However, the concentration of both tocopherols in the liver increases when HS is used in the diet. This means that a higher intake of γ-tocopherol by the cockerels positively influences not only γ-tocopherol, but also α-tocopherol in the liver. The same relationship between the two tocopherols in human blood was presented by Jiang et al. [10]. High doses of α-tocopherol decreased plasma γ-tocopherol, whereas, supplementation with γ-tocopherol increases both tocopherols. Tomazin et al. [33] found different results in chickens. α-Tocopherol or γ-tocopherol were added in equal amounts to a wheat-soy diet with 5% linseed oil. The addition of α-tocopherol increased the α-tocopherol in breast and thigh muscles, but the addition of γ-tocopherol increased only the γ-tocopherol levels. The α-tocopherol concentration in the breast muscles was higher than that in our experiment, and higher than that found in broilers at the same rate by Pompeu et al. [34] based on a meta-analysis. Compared to egg yolk, the muscles of chickens, especially the breast, are a poor source of vitamin E. Although there is a close relationship between vitamin E and the fat content, the differences among tissues and organs are attributed to the genetically determined control of the vitamin E concentration in different categories of poultry, organs and products. There is 30% fat in egg yolk, and the fat content of the liver samples examined in this study was 6.5% and of breast muscle was 1%.

The EF in the diet increased the concentrations of most long-chain n-3 FAs. The n-3 FA most represented in the breast muscle of the cockerels was ALA 18:3. The numerical value of this FA was increased by higher doses of HS (40 or 50 g/kg) and 60 g/kg of EF in the diet, but the increase was not significant. No similar effect was observed in PUFA n-6. This finding is consistent with that of Skiba et al. [35], who stated that deposition efficiency was greater for n-3 than for n-6 FAs. Delta-6-desaturase plays a key role in PUFA biosynthesis, as reported by Brenner [36]. It can be assumed that the effect of flaxseed on the higher production of long-chain n-3 PUFAs was due to the higher activity of delta-5- and delta-6-desaturases and elongases, as described by Garg et al. [37]. 

This study confirms the positive effect of HS in the broiler diet on the instrument-measured tibia deformation and tibia strength originally recorded in hens [11]. Additionally, the combination of HS (40 or 50 g/kg) and EF (60 g/kg) in the diet increased the tibia strength compared to the control. Increased bone strength is of considerable practical importance since fractures are common in all indoor poultry breeding systems. The impetus for measuring the tibia strength was derived from several new studies in the field of human medicine and experiments on rats and bone metabolism [12,13].

## 5. Conclusions

To conclude, the combined dietary addition of HS and EF increased the body weight of the cockerels and decreased the n-6/n-3 PUFA ratio in the breast muscle. The α- and γ-tocopherol contents in the liver, the long-chain n-3 PUFA concentration in the breast muscle and tibia strength were also increased at higher dietary levels of HS and EF. Regarding terms of cockerel performance, meat and bone quality and deposition of α-tocopherol in the liver, the diet combination with 40 g/kg of HS and 60 g/kg of EF was the best among those evaluated.

## Figures and Tables

**Table 1 animals-10-00458-t001:** Nutrient composition of hemp seed and extruded flaxseed.

Item	Hemp Seed	Flaxseed (Extruded) ^1^
Dry matter (g/kg)	960.1	956.4
AME_N_ (MJ/kg)	15.2	12.2
Crude protein (g/kg)	236.5	188.5
Ether extract (g/kg)	300.7	212.6
Crude fibre (g/kg)	270.4	169.8
Calcium (mg/100 g)	177.5	203
Phosphorus (mg/100 g)	1599	838
SFA (mg/100 g)	4510	2844
MUFA (mg/100 g)	6312	3829
PUFA (mg/100 g)	31525	17802
PUFA n-6 (mg/100 g)	24456	4506
PUFA n-3 (mg/100 g)	7068	13296
n-6/n-3	3.460	0.339
α-Tocopherol (mg/kg)	10.20	1.61
γ-Tocopherol (mg/kg)	173.4	61.4

^1^ 70% flaxseed and 30% wheat bran. The flaxseed itself contained 1.06 mg/kg α-tocopherol and 102.7 mg/kg γ-tocopherol. AME_N_ = apparent metabolisable energy, SFA = saturated fatty acid, MUFA = monounsaturated fatty acid, PUFA = polyunsaturated fatty acid.

**Table 2 animals-10-00458-t002:** Composition of broiler diets.

Hemp Seed (g/kg)	0	0	40	30	40	50
Flaxseed (g/kg)	0	60	0	60	60	60
Ingredients (g/kg)						
Wheat	440.0	430.0	430.0	430.0	430.0	430.0
Maize	151.0	142.0	158.8	110.7	107.6	105.5
Soybean meal	300.0	305.0	295.0	315.0	310.0	305.0
Flaxseed (extruded)	-	60.0	-	60.0	60.0	60.0
Wheat bran	40.0	4.0	20.0	-	-	-
Hemp seed	-	-	40.0	30.0	40.0	50.0
Rapeseed oil	30.0	20.0	17.0	15.0	13.0	10.0
Monocalcium phosphate	12.0	12.0	12.0	12.0	12.0	12.0
Limestone	13.0	13.0	13.0	13.0	13.0	13.0
Sodium chloride	2.0	2.0	2.0	2.0	2.0	2.0
Sodium bicarbonate	3.0	3.0	3.0	3.0	3.0	3.0
DL-Methionine	2.0	2.0	2.0	2.0	2.0	2.0
L-Lysine	2.0	2.0	2.2	2.3	2.4	2.5
Vitamin-mineral premix ^1^	5.0	5.0	5.0	5.0	5.0	5.0
Nutrient content (g/kg)						
Dry matter	897.5	889.9	901.3	895.8	898.3	889.4
AME_N_ by calculation, MJ/kg	12.4	12.7	12.6	12.8	12.6	12.8
Crude protein	209.7	210.4	210.9	210.3	210.6	210.4
Ether extract	52.6	52.8	52.4	55.1	54.9	55.4
Crude fibre	40.8	43.5	42.5	42.9	43.8	44.0
Calcium	9.1	9.0	9.3	9.3	9.3	9.5
Phosphorus	5.4	5.6	5.2	5.5	5.7	5.5

^1^ Vitamin-mineral premix provided per kg of diet: retinyl acetate, 3.6 mg; cholecalciferol, 13 µg; niacin, 40 mg; α-Tocopheryl acetate, 30 mg; menadione, 3 mg; thiamine, 3 mg; riboflavin, 5 mg; pyridoxine, 4 mg; cyanocobalamin, 40 µg; calcium pantothenate, 12 mg; biotin 0.15 mg; folic acid, 1.5 mg; choline chloride, 250 mg; ethoxyquin, 100 mg; iron, 50 mg; copper, 12 mg; iodine, 1 mg; manganese, 80 mg; zinc, 60 mg; and selenium 0.2 mg. AME_N_ = apparent metabolisable energy.

**Table 3 animals-10-00458-t003:** Concentrations of fatty acids in the broiler diets (mg/100 g).

Hemp Seed (g/kg)	0	0	40	30	40	50
Flaxseed (g/kg)	0	60	0	60	60	60
C 20:5 n-3	9.13	11.16	13.27	13.25	11.80	13.81
C 22:6 n-3	4.27	4.02	3.30	4.11	3.91	3.90
SFA	908	879	996	1350	1397	1743
MUFA	2596	2061	2091	2638	2712	2902
PUFA	2851	3432	3689	4398	4545	4975
n-3	432	1183	669	1390	1397	1680
n-6	2418	2248	3007	2995	3134	3280
n-6/n-3	5.60	1.90	4.49	2.15	2.24	1.95

SFA = saturated fatty acid, MUFA = monounsaturated fatty acid, PUFA = polyunsaturated fatty acid.

**Table 4 animals-10-00458-t004:** Performance characteristics.

Hemp Seed (g/kg)	0	0	40	30	40	50	SEM	Probability
Flaxseed (g/kg)	0	60	0	60	60	60		
Body weight, 1 D (g)	43.2	43.3	43.3	43.4	43.2	43.3	0.10	NS
Body weight, 14 D (g)	418 ^c^	400 ^d^	427 ^c^	495 ^ab^	511 ^a^	491 ^b^	10.9	<0.001
Body weight, 35 D (g)	2145 ^d^	2254 ^c^	2174 ^cd^	2417 ^ab^	2493 ^a^	2375 ^b^	31.7	<0.001
Feed intake, 1-35 D (g/D)	89.5 ^b^	94.5 ^b^	91.0 ^b^	100.6 ^a^	102.5 ^a^	100.2 ^a^	2.30	<0.05
Feed conversion (kg/kg)	1.52	1.51	1.50	1.50	1.51	1.51	0.030	NS
Mortality (%)	3	2	1	2	2	2	0.4	NS

SEM = standard error of the mean; NS = not significant; ^a^^–^^d^ means in the same row with different superscripts differ significantly.

**Table 5 animals-10-00458-t005:** Concentration of α-tocopherol and γ-tocopherol in the diets, breast meat and liver (mg/kg).

Hemp Seed (g/kg)	0	0	40	30	40	50	SEM	Probability
Flaxseed (g/kg)	0	60	0	60	60	60		
Diet								
α-Tocopherol	55.5	52.0	54.5	53.4	53.9	53.1		
γ-Tocopherol	15.8	15.1	21.5	18.6	18.7	19.0		
Breast meat								
α-Tocopherol	3.85	3.51	3.68	4.06	3.66	3.90	0.146	NS
γ-Tocopherol	0.29	0.31	0.41	0.39	0.37	0.41	0.016	NS
Liver								
α-Tocopherol	15.0 ^abc^	12.6 ^bc^	10.0 ^c^	16.0 ^ab^	18.6 ^a^	19.6 ^a^	0.99	0.017
γ-Tocopherol	1.01 ^ab^	0.77 ^b^	0.75 ^b^	0.87 ^b^	1.10 ^ab^	1.31 ^a^	0.061	0.028

SEM = standard error of the mean; NS = not significant; ^a,b^ means in the same row with different superscripts differ significantly.

**Table 6 animals-10-00458-t006:** Concentration of fatty acids in the breast meat of broilers (mg/100 g).

Hemp Seed (g/kg)	0	0	40	30	40	50	SEM	Probability
Flaxseed (g/kg)	0	60	0	60	60	60		
C 6:0	0.051 ^bc^	0.062 ^ab^	0.066 ^a^	0.066 ^a^	0.057 ^ab^	0.042 ^c^	0.0026	0.011
C 8:0	0.175 ^ab^	0.246 ^a^	0.088 ^c^	0.092 ^c^	0.117 ^bc^	0.108 ^bc^	0.0163	0.007
C 10:0	0.198	0.139	0.112	0.095	0.099	0.119	0.0116	NS
C 12:0	0.768	0.762	0.769	0.737	0.591	0.545	0.0410	NS
C 13:0	0.096	0.107	0.094	0.095	0.074	0.069	0.0066	NS
C 14:0	6.92	4.93	6.10	6.97	4.58	5.86	0.310	NS
C 14:1 n-5	1.698 ^a^	0.806 ^bc^	1.154 ^b^	0.818 ^bc^	0.575 ^c^	1.066 ^b^	0.1003	0.003
C 15:0	1.452	1.054	1.293	1.358	0.972	0.980	0.0748	NS
C 16:0	269	207	265	243	200	242	10.7	NS
C 16:1 n-7	48.5 ^a^	26.4 ^cd^	40.3 ^ab^	32.3 ^bc^	17.6 ^d^	30.4 ^bc^	2.80	0.004
C 17:0	1.66	1.55	1.91	1.96	1.42	1.50	0.103	NS
C 18:0	95.8	88.0	105.1	93.4	87.9	96.8	3.62	NS
C 18:1 n-9	445 ^a^	260 ^cd^	366 ^ab^	359 ^abc^	229 ^d^	320 ^bc^	21.0	0.010
C 18:1 n-7	35.3	26.9	32.1	27.9	21.8	25.9	1.54	NS
C 18:2 n-6 t	0.181 ^a^	0.098 ^b^	0.167 ^a^	0.093 ^b^	0.107 ^b^	0.111 ^b^	0.0097	0.003
C 18:2 n-6	231	177	261	243	196	211	10.7	NS
C 18:3 n-6	4.17	3.54	5.10	5.18	4.33	4.20	0.217	NS
C 18:3 n-3	63.6	53.8	59.4	63.9	77.5	68.9	3.3	NS
C 18:2 (9,11)	1.200	0.896	1.442	1.396	1.343	1.641	0.0820	NS
C 18:2 (10,12)	0.062 ^b^	0.056 ^b^	0.097 ^a^	0.054 ^b^	0.075 ^b^	0.051 ^b^	0.0048	0.011
C 20:0	4.23	4.41	3.79	4.55	4.75	3.59	0.146	NS
C 20:1 n-9	6.86 ^a^	3.74 ^cd^	6.05 ^ab^	5.07 ^bc^	3.26 ^d^	3.96 ^cd^	0.355	0.001
C 20:2 n-6	5.65 ^b^	4.98 ^b^	7.48 ^a^	5.32 ^b^	6.29 ^ab^	5.57 ^b^	0.294	NS
C 21:0	0.480	0.448	0.518	0.441	0.388	0.432	0.0184	NS
C 20:3 n-6	6.48 ^b^	5.68 ^b^	8.68 ^a^	5.64 ^b^	6.88 ^b^	6.09 ^b^	0.326	0.035
C 20:4 n-6	27.8 ^bc^	25.5 ^bcd^	35.4 ^a^	21.7 ^d^	30.3 ^ab^	22.4 ^cd^	1.43	0.019
C 20:3 n-3	2.15 ^c^	4.36 ^b^	2.90 ^c^	4.33 ^bc^	6.20 ^a^	4.92 ^ab^	0.358	<0.001
C 20:4 n-3	0.666 ^bc^	0.900 ^a^	0.803 ^ab^	0.574 ^c^	0.904 ^a^	0.642 ^bc^	0.0414	0.050
C 22:0	0.153 ^d^	0.225 ^cd^	0.344 ^b^	0.260 ^c^	0.422 ^ab^	0.448 ^a^	0.0280	<0.001
C 20:5 n-3	5.34 ^d^	12.11 ^a^	5.72 ^d^	9.25 ^c^	11.86 ^ab^	9.33 ^bc^	0.733	0.002
C 22:1 n-9	0.264 ^b^	0.265 ^b^	0.511 ^a^	0.401 ^ab^	0.414 ^a^	0.394 ^ab^	0.0265	0.021
C 23:0	0.154 ^bc^	0.140 ^c^	0.213 ^ab^	0.224 ^a^	0.153 ^bc^	0.111 ^c^	0.0119	0.012
C 24:0	0.266 ^b^	0.163 ^c^	0.339 ^ab^	0.263 ^b^	0.419 ^a^	0.408 ^a^	0.0253	0.003
C 22:5 n-3	11.3 ^c^	22.1 ^b^	14.1 ^c^	18.4 ^b^	26.5 ^a^	19.3 ^b^	1.33	<0.001
C 24:1 n-9	0.114 ^bc^	0.147 ^b^	0.100 ^c^	0.204 ^a^	0.119 ^bc^	0.090 ^c^	0.0107	0.003
C 22:6 n-3	5.67 ^d^	12.91 ^abc^	10.43 ^bcd^	12.33 ^abc^	16.36 ^a^	10.38 ^cd^	0.960	0.013
SFA	381	309	386	353	302	353	14.6	NS
MUFA	538 ^a^	318 ^cd^	446 ^ab^	426 ^abc^	273 ^d^	382 ^bcd^	25.5	0.011
PUFA	365	324	413	392	385	365	15.3	NS
n-3	88.7 ^c^	106.2 ^bc^	93.4 ^bc^	108.8 ^bc^	139.3 ^a^	113.4 ^ab^	5.80	0.011
n-6	275	217	318	281	244	250	12.3	NS
n-6/n-3	3.11 ^b^	2.03 ^d^	3.41 ^a^	2.58 ^c^	1.75 ^e^	2.21 ^d^	0.144	<0.001

SEM = standard error of the mean; NS = not significant; SFA = saturated fatty acid, MUFA = monounsaturated fatty acid, PUFA = polyunsaturated fatty acid; ^a^^–^^d^ means in the same row with different superscripts differ significantly.

**Table 7 animals-10-00458-t007:** Breaking strength and the mineral content of tibia bones of broilers.

Hemp Seed (g/kg)	0	0	40	30	40	50	SEM	Probability
Flaxseed (g/kg)	0	60	0	60	60	60		
Breaking strength (N)	297.3 ^b^	304.6 ^ab^	346.3 ^a^	339.0 ^ab^	359.2 ^a^	358.3 ^a^	8.62	<0.001
Deformation (mm)	6.32 ^a^	5.40 ^b^	5.36 ^b^	5.50 ^b^	5.24 ^b^	5.50 ^b^	0.17	<0.001
Ash (g/kg)	486.4	481.7	478.6	472.9	485.8	469.1	2.15	NS
Calcium (g/kg ash)	375	369	377	363	364	369	2.1	NS
Phosphorus (g/kg ash)	178	179	179	177	178	177	0.5	NS
Magnesium (g/kg ash)	8.62	8.76	8.91	8.75	9.00	9.32	0.093	NS

SEM = standard error of the mean; NS = not significant; ^a,b^ means in the same row with different superscripts differ significantly.

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
