# Peer review of "Effects of Dietary Hemp Seed and Flaxseed on Growth Performance, Meat Fatty Acid Compositions, Liver Tocopherol Concentration and Bone Strength of Cockerels"

_animals, 2020, doi:10.3390/ani10030458_

Round 1

Reviewer 1 Report

Re-review of the manuscript.  Authors made the requested changes and the manuscript should be accepted for publication.

Author Response

Re-review of the manuscript.  Authors made the requested changes and the manuscript should be accepted for publication.

Answer: Dear reviewer, we would like to thank you again for the comments which were beneficial for the final quality of this article and for your decision.

Reviewer 2 Report

The article is substantially improved. However, some points should be corrected:

L31: "replicates" instead of "repetitions"

L33: "...which lasted 35 days..."

L45: "dietary supplementation" instead of "diet combination"

L57: "...tocopherols (γ-, α-, δ-,  and β-) in seeds..."

L58: "Due to" instead of "Because of" 

L84-86: Please rephrase

L87: "...fed a diet supplemented with linseed oil..."

L105, 106: group

L115: "...in Tables 1, 2 and 3."

L116: "adjusting" instead of "varying"

L246: "In contrast to the majority of the published data..."

L250: Polyphenols are antinutritional agents?

L259: "in the background"?

L264-265: Please rephrase

L273-274: Please rephrase

L276: Please rephrase

L279-280: Please merge sentences

Author Response

Answer: Dear reviewer, we would like to thank you again for the comments of the article and requirements for its correction. All of your requirements were beneficial for the final quality of this article. We have accepted all of your requirements and adjusted the text according to them. Adjustments of the text were made in accordance with the requirements of another reviewer. All changes in the manuscript are marked in yellow.

The article is substantially improved. However, some points should be corrected:

L31: "replicates" instead of "repetitions"

Answer: The word was corrected.

L33: "...which lasted 35 days..."

Answer: The sentence was corrected.

L45: "dietary supplementation" instead of "diet combination"

Answer: The sentence was corrected.

L57: "...tocopherols (γ-, α-, δ-,  and β-) in seeds..."

Answer: The sentence was corrected.

L58: "Due to" instead of "Because of" 

Answer: The sentence was corrected.

L84-86: Please rephrase

Answer: The sentence was rewritten.

L87: "...fed a diet supplemented with linseed oil..."

Answer: The sentence was corrected.

L105, 106: group

Answer: The word was corrected.

L115: "...in Tables 1, 2 and 3."

Answer: The sentence remained unchanged, because the concentration of α-tocopherol and γ-tocopherol in the diets is mentioned in the Table 5.

L116: "adjusting" instead of "varying"

Answer: The word was corrected.

L246: "In contrast to the majority of the published data..."

Answer: The sentence was corrected.

L250: Polyphenols are antinutritional agents?

Answer: Yes, polyphenols can act as antinutritional agents (e.g. tannins) as well as nutritionally beneficial substances. It depends on the source and many nutritional factors.

L259: "in the background"?

Answer: The sentence was reformulated.

L264-265: Please rephrase

Answer: The sentence was explained.

L273-274: Please rephrase

Answer: The sentence was rephased.

L276: Please rephrase

Answer: The sentence was rephased.

L279-280: Please merge sentences

Answer: The sentences were merged.

Reviewer 3 Report

The authors have made a number of changes to the manuscript and have addressed some of the points raised in the critique of the original submission (as Animals-668649). The fundamental problems relating to the way in which the feeds were formulated, with changes in the proportions of several ingredients (Table 1) remain, so it is not possible to attribute any observed treatment effects to feed supplementation with hemp- and flax-seed, per se.

There are several aspects of the presentation that require attention, and there is scope for improvement of the manuscript.

The title is a bit cumbersome, the hierarchy in presentation of the studied metrics is not ideal and the use of the word ‘performance’ is a bit vague. It might be better to specify ‘growth metrics’, rather than use ‘performance’, and to present this first in the list: Effects of dietary hempseed and flaxseed on growth metrics, meat fatty acid compositions, liver tocopherol concentration and bone strength of cockerels.

The keywords have been improved. Expand the keyword covering fatty acids to include wording: n-3 polyunsaturated fatty acid (n-3 PUFA).

The simple summary is difficult to read and it probably contains too little quantitative information. Starting the simple summary with a reference to human health may confuse readers; it might be best to delete the sentence on lines 14-15. The wording ‘performance characteristics’ is too vague; be specific and provide some numerical values about the findings. Note: On line 16, broiler has not been replaced by cockerel.

In the Abstract be more specific about ‘performance’ (line 29), include information about dietary energy and protein concentrations (line 32), give information about the lipid (oil) source used in the control diet (line 33), write out polyunsaturated fatty acids (PUFA) in full (line 43) and delete the wording ‘in broiler chickens’ from line 45. On lines 38-39 it is unclear whether all HS + EF combinations gave similar effects; make this clear. The sentence on lines 39-41 is vague, and somewhat confusing given the lack of clarity in the preceding sentence.

There is some cumbersome sentence structure in the Introduction, so work on the language is required. For example, on line 54 replace ‘fat’ with ‘oil’ and delete the word ‘individual’, and on line 55 replace the wording ‘identified these substances in’ with ‘examined’.

On line 65, EF is introduced without being explained; define acronyms and abbreviations when first mentioned in the main text. Some background information should probably be given about flaxseed (EF) along the same lines as given for HS.

On line 66, the wording ‘broiler chicken’ is used; it might be better to use the broader term ‘poultry’, especially as ‘poultry feed’ is used on line 67, and ‘poultry performance’ is used on line 68.

There is an abrupt transition on line 68, a sudden switch from ‘poultry feed’ to ‘broiler diet’, and it is unclear what the sentence on lines 68-69 means. There then follows a passage of text in which there are switches between ‘broiler’ and ‘chickens’. The text on lines 67-77 needs to be re-worked.

On line 80 amend the text as follows ‘….total FAs [13], and the ratio of LA (n-6) to ALA (n-3) is approximately 3.3’.

The presentation of the attributes of flaxseed should probably be moved to earlier in the Introduction; immediately following the presentation of hempseed. Note: the formal name of flax has not been included, whereas that of hemp has.

The text covering lines 82-90 has a cumbersome structure, and the sudden introduction of ‘poultry’ as a source of EPA and DHA (in human nutrition?) disrupts the flow of the text.

The text on line 91 et seq should probably be incorporated into the general presentation of the characteristics and properties of hempseed given at the start of the Introduction.

The M & Ms has been modified to provide more information about the reasons for opting for particular feed formulations and there are more details given about the sampling and several of the protocols.

On line 104 specify that rapeseed was used as the oil source in the control diet: ‘The control diet had rapeseed oil as the oil source, and did not…..’

On line 105 it is not correct to use the word ‘supplemented’ because this implies that these came as additional oil sources without rapeseed oil being reduced. Replace the word ‘supplemented’ with ‘contained’ and add a brief phrase along the lines of ‘and the dietary level of rapeseed oil was reduced in comparison with the control’.

The text on lines 107-119 has a cumbersome and disjointed structure, and this makes it difficult to read. It should probably be mentioned that the diets were isoenergetic and isonitrogenous very early in the description of the M & Ms, e.g. on line 104, and the information about energy and protein concentrations given in brackets.

Table 2 should probably be re-numbered Table 1 and reference made to this table on line 104: ‘content of HS and extruded flaxseed (EF) (Table 1) in the diet (Table 2)’. Table 1 is the composition HS and EF, whereas table 2 is the overview of diet formulations and compostion.

It is the pen, and not individual birds, that is the experimental unit so n = 3 for all data and statistical analysis. The use of data for individual birds, e.g. table 4, is a case of pseudo-replication and the data will have to be re-worked. In table 4 give the data in terms of average weights of the birds in each pen (n = 3) and make the statistical analyses using pen means as the basis.

The sentence on lines 127-128 is unclear; provide specific information about which behavioral observations were involved and the criteria used for assessing health status.

The inclusion of the words ‘each component’ in the sentence on lines 153-154 is confusing and makes the meaning difficult to understand; what is ‘each component’? The sentence could be simplified and made clearer: The dry matter of the diets, HS and EF was determined by drying to constant weight at 105 C in an oven…..

On line 157 it is incorrect to use the words ‘dietary components’, because the ‘dietary components’ (e.g. wheat, maize, soybean meal etc) were not analyzed. The sentence should be re-worded: Dry homogenized diets, HS and EF were heated to 550 C (give name of the muffle oven/furnace used) and the mineral ash dissolved in 3 M hydrochloric acid.

In the text (lines 158 and 160) P and Ca are used, but in tables 1, 2 and 7 phosphorus and calcium are written out in full; write out in full in the text.    

On line 187, Mg is used, but in table 7 magnesium is written out in full; write out in full in the text.

On lines 162, 163, 165 and 180, n is given as 15, which implies that individual birds were used as the experimental unit; this is pseudo-replication and pen means (n = 3) should be used in the analyses.

Given the problems with the presentation (e.g. in the general structure, description of the M & Ms and data analysis) it is premature to make comments about the Results and Discussion sections of the manuscript.

Author Response

Answer: Dear reviewer, we would like to thank you again for the comments of the article and requirements for its correction. All of your requirements were beneficial for the final quality of this article. We have adjusted the text according to your requirements. Adjustments of the text were made in accordance with the requirements of another reviewer. All changes in the manuscript are marked in yellow.

The authors have made a number of changes to the manuscript and have addressed some of the points raised in the critique of the original submission (as Animals-668649). The fundamental problems relating to the way in which the feeds were formulated, with changes in the proportions of several ingredients (Table 1) remain, so it is not possible to attribute any observed treatment effects to feed supplementation with hemp- and flax-seed, per se.

Answer: Changes in the proportion of some ingredients in feed mixtures were necessary to maintain their isoenergetic and isonitrogenic principles.The dietary concentrations of SFA, MUFA and PUFA demonstrate the accuracy of this procedure. The concentrations of SFA, MUFA and PUFA increased in accordance with the increasing dietary addition of hemp seed.

There are several aspects of the presentation that require attention, and there is scope for improvement of the manuscript.

The title is a bit cumbersome, the hierarchy in presentation of the studied metrics is not ideal and the use of the word ‘performance’ is a bit vague. It might be better to specify ‘growth metrics’, rather than use ‘performance’, and to present this first in the list: Effects of dietary hempseed and flaxseed on growth metrics, meat fatty acid compositions, liver tocopherol concentration and bone strength of cockerels.

Answer: The title was changed according to your design.

The keywords have been improved. Expand the keyword covering fatty acids to include wording: n-3 polyunsaturated fatty acid (n-3 PUFA).

 Answer: The keyword was rewritten.

The simple summary is difficult to read and it probably contains too little quantitative information. Starting the simple summary with a reference to human health may confuse readers; it might be best to delete the sentence on lines 14-15. The wording ‘performance characteristics’ is too vague; be specific and provide some numerical values about the findings. Note: On line 16, broiler has not been replaced by cockerel.

Answer: The simple summary was modified and some numerical values about the findings were added.

In the Abstract be more specific about ‘performance’ (line 29), include information about dietary energy and protein concentrations (line 32), give information about the lipid (oil) source used in the control diet (line 33), write out polyunsaturated fatty acids (PUFA) in full (line 43) and delete the wording ‘in broiler chickens’ from line 45. On lines 38-39 it is unclear whether all HS + EF combinations gave similar effects; make this clear. The sentence on lines 39-41 is vague, and somewhat confusing given the lack of clarity in the preceding sentence.

Answer: The abstract was modified, the dietary energy and protein concentrations and lipid source were added, The sentences on lines 38-41 were corrected.

There is some cumbersome sentence structure in the Introduction, so work on the language is required. For example, on line 54 replace ‘fat’ with ‘oil’ and delete the word ‘individual’, and on line 55 replace the wording ‘identified these substances in’ with ‘examined’.

Answer: The sentence was corrected according to your remark.

On line 65, EF is introduced without being explained; define acronyms and abbreviations when first mentioned in the main text. Some background information should probably be given about flaxseed (EF) along the same lines as given for HS.

Answer: The abreviation was defined. Some background information about flax seed was added.

On line 66, the wording ‘broiler chicken’ is used; it might be better to use the broader term ‘poultry’, especially as ‘poultry feed’ is used on line 67, and ‘poultry performance’ is used on line 68.

Answer: The term was changed.

There is an abrupt transition on line 68, a sudden switch from ‘poultry feed’ to ‘broiler diet’, and it is unclear what the sentence on lines 68-69 means. There then follows a passage of text in which there are switches between ‘broiler’ and ‘chickens’. The text on lines 67-77 needs to be re-worked.

Answer: The text was re-worked.

On line 80 amend the text as follows ‘….total FAs [13], and the ratio of LA (n-6) to ALA (n-3) is approximately 3.3’.

Answer: The text was changed.

The presentation of the attributes of flaxseed should probably be moved to earlier in the Introduction; immediately following the presentation of hempseed. Note: the formal name of flax has not been included, whereas that of hemp has.

Answer: The information about flax seed was added.

The text covering lines 82-90 has a cumbersome structure, and the sudden introduction of ‘poultry’ as a source of EPA and DHA (in human nutrition?) disrupts the flow of the text.

Answer: The text on lines 82-90 was rewritten.

The text on line 91 et seq should probably be incorporated into the general presentation of the characteristics and properties of hempseed given at the start of the Introduction.

 Answer: The paragraph was relocated.

The M & Ms has been modified to provide more information about the reasons for opting for particular feed formulations and there are more details given about the sampling and several of the protocols.

On line 104 specify that rapeseed was used as the oil source in the control diet: ‘The control diet had rapeseed oil as the oil source, and did not…..’

 Answer: The sentence was enriched according to the remark.

On line 105 it is not correct to use the word ‘supplemented’ because this implies that these came as additional oil sources without rapeseed oil being reduced. Replace the word ‘supplemented’ with ‘contained’ and add a brief phrase along the lines of ‘and the dietary level of rapeseed oil was reduced in comparison with the control’.

Answer: The text was rewritten.

The text on lines 107-119 has a cumbersome and disjointed structure, and this makes it difficult to read. It should probably be mentioned that the diets were isoenergetic and isonitrogenous very early in the description of the M & Ms, e.g. on line 104, and the information about energy and protein concentrations given in brackets.

Answer:The sentences were relocated and enriched.

Table 2 should probably be re-numbered Table 1 and reference made to this table on line 104: ‘content of HS and extruded flaxseed (EF) (Table 1) in the diet (Table 2)’. Table 1 is the composition HS and EF, whereas table 2 is the overview of diet formulations and compostion.

Answer: The tables were re-numbered and the reference was added.

It is the pen, and not individual birds, that is the experimental unit so n = 3 for all data and statistical analysis. The use of data for individual birds, e.g. table 4, is a case of pseudo-replication and the data will have to be re-worked. In table 4 give the data in terms of average weights of the birds in each pen (n = 3) and make the statistical analyses using pen means as the basis.

Answer: We used n=90 in body weight, n=3 in other performance characteristics and n=15 in meat and bone quality characteristics (because at 35 days of age, 15 cockerels (5 cockerels from each pen, 5 x 3) with an average body weight were selected from each dietary treatment group and slaughtered). We think that in terms of statistics, the reducing of “n” number is not reasonable. We have followed our articles that have been published in leading magazines over the last decades. The same methodological procedure has been previously successfully used and published, e.g. Skřivan M., Marounek M., Englmaierová M., Skřivanová E.: Influence of dietary vitamin C and selenium, alone and in combination, on the composition and oxidative stability of meat of broilers. Food Chemistry, 2012, 130(3), 660 – 664. This comment has occurred for the first time. Thank you for it. We'll take it into account in our next work.

The sentence on lines 127-128 is unclear; provide specific information about which behavioral observations were involved and the criteria used for assessing health status.

Answer: The text was enriched in information about criteria used for assessing health status.

The inclusion of the words ‘each component’ in the sentence on lines 153-154 is confusing and makes the meaning difficult to understand; what is ‘each component’? The sentence could be simplified and made clearer: The dry matter of the diets, HS and EF was determined by drying to constant weight at 105 C in an oven…..

Answer: The sentence was reformulated.

On line 157 it is incorrect to use the words ‘dietary components’, because the ‘dietary components’ (e.g. wheat, maize, soybean meal etc) were not analyzed. The sentence should be re-worded: Dry homogenized diets, HS and EF were heated to 550 C (give name of the muffle oven/furnace used) and the mineral ash dissolved in 3 M hydrochloric acid.

Answer: The sentence was reformulated.

In the text (lines 158 and 160) P and Ca are used, but in tables 1, 2 and 7 phosphorus and calcium are written out in full; write out in full in the text.    

Answer: P and Ca were written in full in the text.

On line 187, Mg is used, but in table 7 magnesium is written out in full; write out in full in the text.

Answer: Mg was written in full in the text.

On lines 162, 163, 165 and 180, n is given as 15, which implies that individual birds were used as the experimental unit; this is pseudo-replication and pen means (n = 3) should be used in the analyses.

Answer: We think that in terms of statistics, the reducing of “n” number is not reasonable. We have followed our articles that have been published in leading magazines over the last decades. The same methodological procedure has been previously successfully used and published, e.g. Skřivan M., Marounek M., Englmaierová M., Skřivanová E.: Influence of dietary vitamin C and selenium, alone and in combination, on the composition and oxidative stability of meat of broilers. Food Chemistry, 2012, 130(3), 660 – 664. This comment has occurred for the first time. Thank you for it. We'll take it into account in our next work.

Given the problems with the presentation (e.g. in the general structure, description of the M & Ms and data analysis) it is premature to make comments about the Results and Discussion sections of the manuscript.

Round 2

Reviewer 3 Report

The authors have made a number of changes to the manuscript, including some re-structuring, correction of language and the giving of some additional information. They have not, however, addressed the point relating to pseudo-replication in the data analysis.

It is the pen, and not individual birds, that is the experimental unit so n = 3 for all data and statistical analysis. The use of data for individual birds, e.g. table 4, is a case of pseudo-replication and the data will have to be re-worked. In table 4 give the data in terms of average weights of the birds in each pen (n = 3) and make the statistical analyses using pen means as the basis.

The same applies to other analyses where data for individual birds has been used.

The manuscript should not be published with flawed data analysis, so the authors are strongly advised to make the effort to re-analyze their data in a correct manner. It seems unlikely that a re-analysis of the data will lead to any large-scale changes in the overall conclusions, but this must be confirmed.

Author Response

Dear reviewer,

we would like to thank you again for the comments of the article and requirements for its correction. All of your requirements were beneficial for the final quality of this article. We have accepted all of your requirements and adjusted the text according to them. Data were statistically recalculated; the experimental unit is n = 3. All changes in the manuscript are marked in yellow.

On behalf of authors,

Prof. Miloš Skřivan